# Geriatrics as a specialty: Insights from Mexican medical students' decision-making process

Carolina Gómez-Moreno[1], Tania Vives-Varela[2], Alfonso Gulias-Herrero[3], José Alberto Ávila-Funes [4], Ayari Pérez-Méndez[1], Melchor Sánchez-Mendiola[5], Carlos Gutiérrez-Cirlos [3,5] *

1 Department of Emergency Medicine, National Institute of Medical Sciences and Nutrition Salvador Zubirán, Mexico City, Mexico, 2 Faculty of Medicine, Department of Medical Education, National Autonomous University of Mexico, Mexico City, Mexico, 3 Internal Medicine Department, National Institute of Medical Sciences and Nutrition Salvador Zubirán, Mexico City, Mexico, 4 Medical Education Department, National Institute of Medical Sciences and Nutrition Salvador Zubirán, Mexico City, Mexico, 5 Faculty of Medicine, National Autonomous University of Mexico, Mexico City, Mexico

☯ These authors contributed equally to this work.
* alfonso.gutierrezc@incmnsz.mx cirlos@hotmail.com

**Data Availability Statement:** For long-term availability date, we provide the durable points of

## Abstract

### Main objective

To identify the elements associated with the decision to select or reject geriatrics as a medical specialty in Mexico.

### Methods

Qualitative, with the focus group technique to obtain data. Two groups included students who rejected geriatrics, regardless of the specialty they chose, and two groups included students who chose geriatrics, interviews via Zoom were conducted. Data was analyzed qualitatively based on grounded theory.

### Results

Thirty-four students were included in the final sample (52% women), mainly from the Central region of the country (44%). Students were mainly in their last year of medical school (97%). Previous experiences with older patients and older close relatives, both positive and negative, were important in choosing the specialty of geriatrics for most students. In contrast, personality, and interpretation of the experiences as good or bad was the main differentiator between groups. For those who did not choose geriatrics, the main factor was the lack of knowledge regarding contents, aims, and professional opportunities for geriatricians. Previous experiences with geriatric care influenced positively towards choosing geriatrics as a medical specialty. High-quality, structured exposure to geriatrics should be encouraged among medical students in order to increase knowledge and eventually interest in the specialty.

contact information: Sergio Hernández Jiménez President of the Ethics and Human Research Committees, National Institute of Medical Sciences and Nutrition Salvador Zubirán. Link:https://www.incmnsz.mx/opencms/contenido/investigacion/comiteEtica/com_etica_inv.html Email: sergio.hernandezj@incmnsz.mx Telephone: + 52 55 54 87 0900 ext. 6101 Carlos A. Aguilar Salinas Research Director, National Institute of Medical Sciences and Nutrition Salvador Zubirán Link:https://www.incmnsz.mx/opencms/contenido/investigacion/director.html Email: carlos.aguilars@incmnsz.mx Telephone: +52 55 54 87 0900 ext. 6107, 6109.

**Funding:** The author(s) received no specific funding for this work.

**Competing interests:** The authors have declared that no competing interests exist.

## Introduction

Nowadays, despite the fact that the global number of physicians has increased, some specialties have experienced insufficient interest towards them and a low and even decreasing number of doctors that select those areas [1–3]. Some of these less popular specialties include genetics, epidemiology, public health, and geriatrics [4].

The lack of interest in geriatrics is concerned for governments and health institutions because of the worldwide aging phenomenon. For instance, in Mexico it is expected that about 17% of the population would be 65 years or more by 2050, such an increase being consistent around the global population [5]. However, regardless of the growing need for geriatricians, geriatric's training programs report up to a 50% of vacancy for first year residents in programs in the USA [6].

The natural history of recruitment towards a new field within medicine is one of a hesitant beginning followed by flourishing training programs over a two-decade period. Unfortunately, in geriatrics this natural history has experienced a delay without a clear explanation of why this is happening [7].

Specialty selection is a pivotal point in a doctor's career and is a complex process that has proved hard to analyze since it's influenced not only by the student's goals, but by the expectation of the centers offering the programs and the competition for available spots in any given residency program [1,3]. Publications have attempted to explain the process of specialty selection with interesting results. A review that included numerous studies from 25 different countries, 18 high income countries, 2 upper middle income countries, 4 lower middle income countries and 1 low income country [8], showed that the elements influencing decisions regarding medical specialties were: academic interest, the students' competencies, the type of patients, mentors or teachers that had impact in the students, the work burden associated with each specialty, the training duration but also the prestige associated with the specialty, counseling from family and friends and amount of student debt [4]. Even though this study gives valuable insights, some elements might not apply in different settings.

Furthermore, when studying a phenomenon such as specialty selection, the aim is to understand what the participants are thinking and doing at that moment [9], therefore traditional quantitative studies may not be the best approach to gain a deeper understanding of the process. To achieve this, a qualitative approach provides an appropriate framework to analyze this dynamic event [9]. This understanding is of particular interest for medical areas that, like geriatrics, have apparently struggled to gain popularity but are increasingly relevant due to the health care needs of an ever-aging global population.

We conducted a qualitative study to identify the factors, expectations, ideas, and opinions associated with choosing or rejecting geriatrics specialty among Mexican medical students.

## Materials and methods

This was a qualitative study with a phenomenological orientation using focus groups. The phenomenological approach, assumes the analysis of profound elements of human experience, its objective is to understand the lived experiences in their complexity, searching to gain awareness and meaning around the object of study [10]. Given that specialty selection is a unique and complex phenomenon in the lives of medical students, a phenomenological approach provides the necessary framework for a deeper understanding of the process from the perspective of those who have lived it [11]. We chose to use focus groups since the dynamic achieved with them allows us to examine in detail the experiences in the context of health care, as well as being an effective method to explore the attitudes and needs of the participants [12–14].

## Ethical considerations

The protocol was approved by the ethics and research committees of the National Institute of Medical Sciences and Nutrition Salvador Zubirán approval number. PMDCMOS/CE3/03/ 2021 and by the ethics counsel of the Masters and Doctorate Program of Medical and Odonatological Sciences of the National Autonomous University of Mexico approval: CONBIOÉTICA-09CEI-011-20160627.

All participants gave their consent to participate and to be recorded which were collected and recorded before starting the interviews. Students were free to keep the camera on their devices open or closed.

## Study population and setting

Our target population were medical students willing to pursue a medical specialty in Mexico and were going through the selection process, either Mexican or foreigners living and studying medicine in Mexico. This was a qualitative study with the use of focus groups, we used consecutive, convenience sampling. As recommended in the literature we aimed to construct at least 2 groups per point of view (geriatrics vs no-geriatrics) with between 8 and 12 participants per group, so our estimated sample was of at least 32 and as many as 48 students [15], however if data saturation was not achieved after the first four groups, further enrolment was planned and could have continued, however, since we did achieve data saturation after the four interviews to the initial focus groups, no more enrolment was required. To graduate as a physician in Mexico, it is mandatory to go through a clinical internship with a one-year duration and afterwards another year in social service. This last year was a governmental program in which medical students help with vulnerable populations in different areas of the country, it is mandatory for undergraduate students. These two final years of medical school are the period in which interested students apply to the medical specialty selection process; therefore, our sample included students during this period in their career.

Students were invited to participate and afterwards recruited consecutively via social media. We posted an invitation in the personal social media accounts of one of the researchers (CGM) in Facebook® and Twitter®, considering methodological and ethical recommendations previously published [16]. Both posts included complementary contact information such as the mail address and Instagram® account of the researcher and were enabled to be shared by the researcher's contacts. Interested participants contacted the researcher through mail, direct messaging in Twitter® or Facebook® messenger.

Forty students reached out after the invitation. Six students dropped out mainly due to conflicting schedules. Four focus groups were used to identify and explore the participants' thoughts regarding their specialty selection. Two groups included students that were convinced to select geriatrics and two groups included students that would not study geriatrics regardless of their specialty selection. The final number of groups was defined as we reached saturation point from the interviews.

The groups were assembled trying to balance by sex to achieve a 50–50% or 40–60% ratio between women and men. We also tried to include 1 or 2 participants from each of the 5 educational regions that are used by the Mexican Public Education Ministry, and both participants from public and private universities were included. All participants in the focus groups contributed to the interview. Characteristics of the groups' distribution are available in Fig 1.

## Data collection

Two instruments were developed, one for the no-geriatrics groups and one for the geriatrics groups (Tables 1 and 2). Interviews were done from October 2021 to March 2022. Both instruments were based on the work published by Gutiérrez-Cirlos et al. in a study evaluating

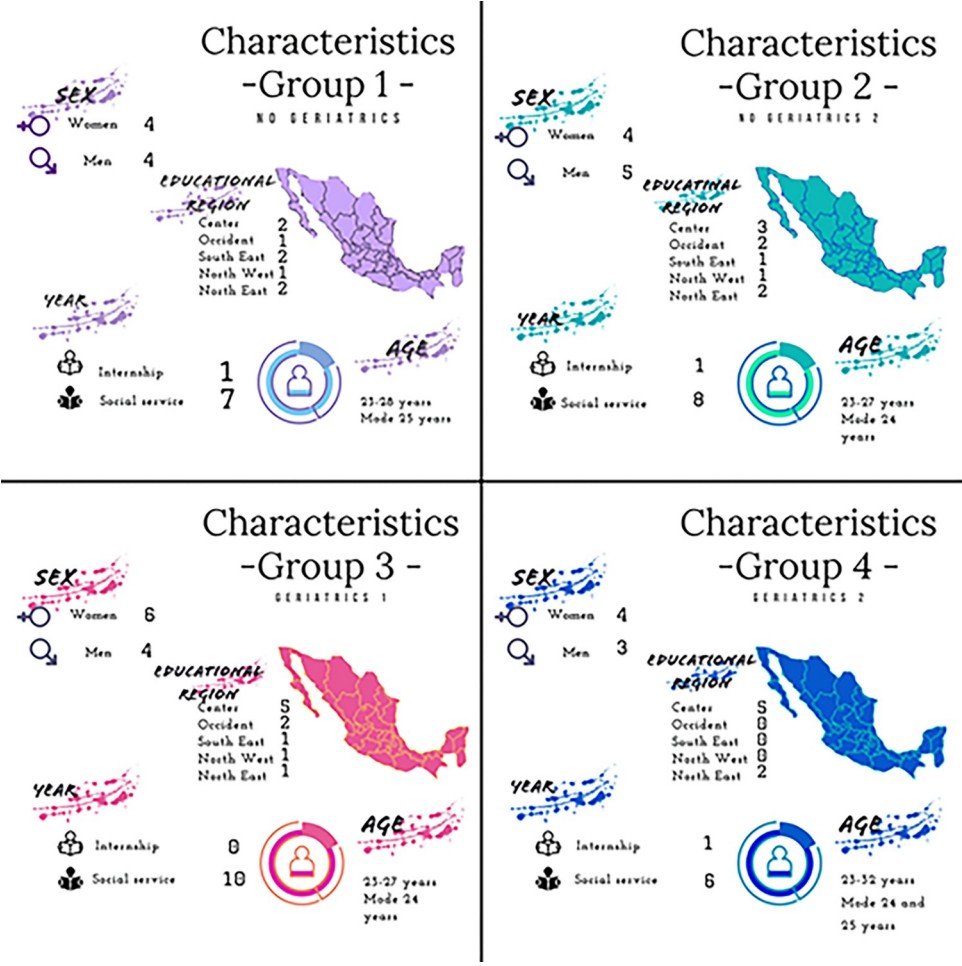

**Fig 1. All focus groups.** Groups 1 and 2 correspond to students that reject geriatrics, group 3 and 4 correspond to students that selected geriatrics.

specialty selection [17]. The instruments and interviews were developed and conducted in Spanish, and the figures and testimonies presented here were translated for publication purposes. No pilot tests were conducted since the questions were already validated in the aforementioned study. For the interviews, the platform Zoom® was used, each focus group was recorded, an audio backup recording was made using a cellphone. Focus groups durations were 55 minutes, 75 minutes, 61 minutes, and 50 minutes respectively for groups 1, 2, 3 and 4.

The first focus group was coordinated by two researchers, TV and CGM, and groups 2, 3 and 4 were coordinated solely by CGM. After the second focus group for geriatrics and no-geriatrics students, data saturation was discussed and confirmed by TV and CGM, therefore no further recruitment was required.

The transcriptions were reviewed by at least one of the students that participated in each group to ensure validity and reliability of the data, they read the transcription and confirmed that the content reflected reliably what was commented on in their respective interviews.

### Data analysis

At a first level of analysis, the testimonies were coded based on the categories of the work of Gutiérrez-Cirlos et al. producing the main category tree [17] (Fig 2). For optimal inter-

**Table 1. Instrument for focus groups with "no-geriatrics" students.**

| |
|---|
| 1. Did you ever consider choosing geriatrics as a specialty? |
| 2. What would you say was the MAIN reason you wouldn't choose geriatrics?<br> Besides this main reason, are there any other elements that you considered for not choosing geriatrics? Do you recall these elements? Would you consider that these elements were determinant in your decision? |
| 3. When in your career did you make the decision of definitely not studying geriatrics? |
| 4. Was there a person or person that influenced your decision to reject geriatrics? |
| 5. Which components of your personality do you think were important in rejecting geriatrics? |
| 6. Did the lifestyle associated with the specialty influence your decision not to study geriatrics?<br> 1. Did you consider elements such as economic remuneration and social prestige to reject geriatrics? |
| 7. What importance did the medical school curriculum have in not choosing geriatrics? |
| 8. What was the influence of the hospitals of your clinical rotations in your decision to not study geriatrics?<br> 1. Hospitals where you had already attended and/or<br> 2. Hospitals where you would like to study in the future |
| 9. Do you consider that you know enough about geriatrics in order to be absolutely sure that you will NEVER study it as a specialty?<br> 1. What information is missing or is it biased in order to make an informed and clear decision regarding the specialty?<br> 2. Do you have any examples regarding this? |
| 10. What else do you consider should be asked to evaluate the factors that influence specialty selection? |

subjectivity agreement among coders, three researchers (CGM, TV, CC) coded the transcripts of the focus groups separately, until unification of the coding criteria was achieved [26]. Themes/categories were identified in advance, using the work by Gutiérrez-Cirlos et al as a guide [17]; new themes were added according to the findings in the interviews.

The quotes were marked as follows: G: geriatrics group; NG: non-geriatrics group, the date in which the interview was conducted and M: masculine; F: feminine participant, for example: NG-17.09.2021_M.

A second level of analysis was performed to generate general explanatory categories that accounted for the elements that influence the choice or rejection of geriatrics as a specialty, for this, a triangulation with the results from each category obtained in the initial codification and with the literature from the specific subject was conducted [18], in order to generate general

**Table 2. Instrument for focus groups with "geriatrics" students.**

| |
|---|
| 01. What elements did you consider to be the most important in your specialty selection?<br> a. Would you consider that these elements were determinant in your decision?<br> b. If such elements were not determinant, what kind of influence did they have? |
| 02. When in your career did you make the decision to study geriatrics? |
| 03. Was there a person or person that influenced your decision to choose geriatrics? |
| 04. What elements of your personality do you think were important in your decision? |
| 05. What characteristics regarding lifestyle, financial remuneration and prestige of the specialty motivated you to choose geriatrics? |
| 06. How important was the university curriculum in your selection? |
| 07. What was the influence of the hospitals of your clinical rotations in your decision to study geriatrics?<br> a. Hospitals where you had already attended and/or<br> b. Hospitals where you would like to study in the future |
| 08. Did bullying of older students and/or your classmates have any impact? |
| 09. What information is missing or is it biased in order to make an informed and clear decision regarding the specialty? |
| 10. If you hadn't chosen geriatrics, what other specialty would you have selected? |
| 11. What else do you consider should be asked to evaluate the factors that influence specialty selection? |

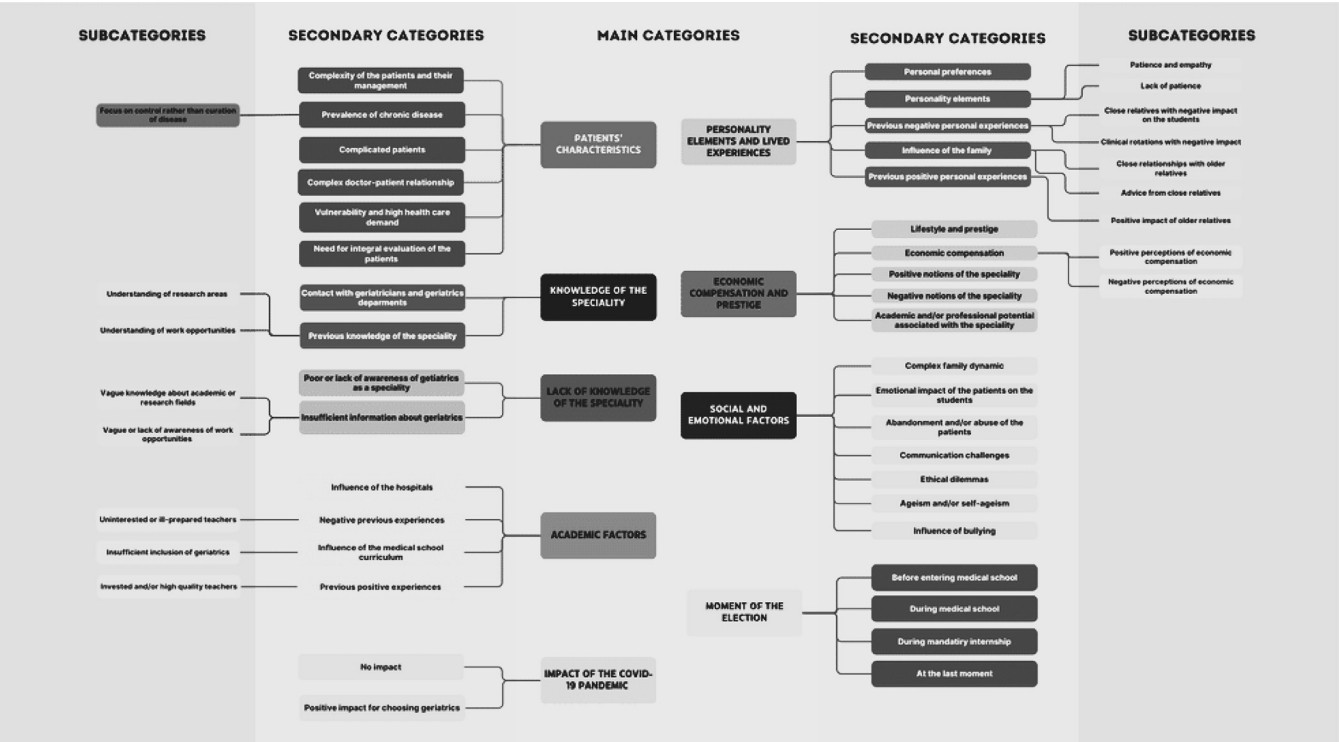

**Fig 2. First level analysis category tree.** Category tree obtained from the students' statements during the interviews, first tier analysis.

explicative categories that considered the elements that weigh in the selection or rejection of geriatrics as a specialty. This process is exemplified in Fig 3, in this representation, it can be observed the grouping of the categories obtained from the interviews that formed the main category tree, into central themes, a curved line represents the inclusion of the categories into its corresponding theme. These central themes had an impact in the decision-making process which is qualitatively represented as curved lines, it should be noted that each central theme has two lines emerging from their title: one that connects with "geriatrics" and one that connects with "no-geriatrics" and therefore to the decision to select or reject geriatrics respectively. The thicker the line, the higher number of quotes identified about the theme, for instance, for the central theme "Patients' characteristics" both lines are of similar thickness, whereas for the central theme "Understanding of the specialty" the line that connects with "no-geriatrics" is thicker than the line that connects with "geriatrics". In this representation, we can also qualitatively see, that the central theme with the highest volume of quotes is "Sociocultural elements" as its connecting lines are significantly thicker compared to the lines emerging from the rest of the central themes.

Credibility was achieved by approaching the subject in preparation of the study, as well as through the literature review needed for the background and the triangulation, since our findings resonated with some previous knowledge, as well as providing new information. Data collection took place over a six-month period, allowing for an iterative process of analysis and adjustment. Confirmability was addressed by creating an environment of openness, trust, and dialogue within the focus group as well as through a joint and critical analysis by the research team on methodological decisions throughout the research process [19,20], additionally the researchers' characteristics involved in the coding process were carefully considered:

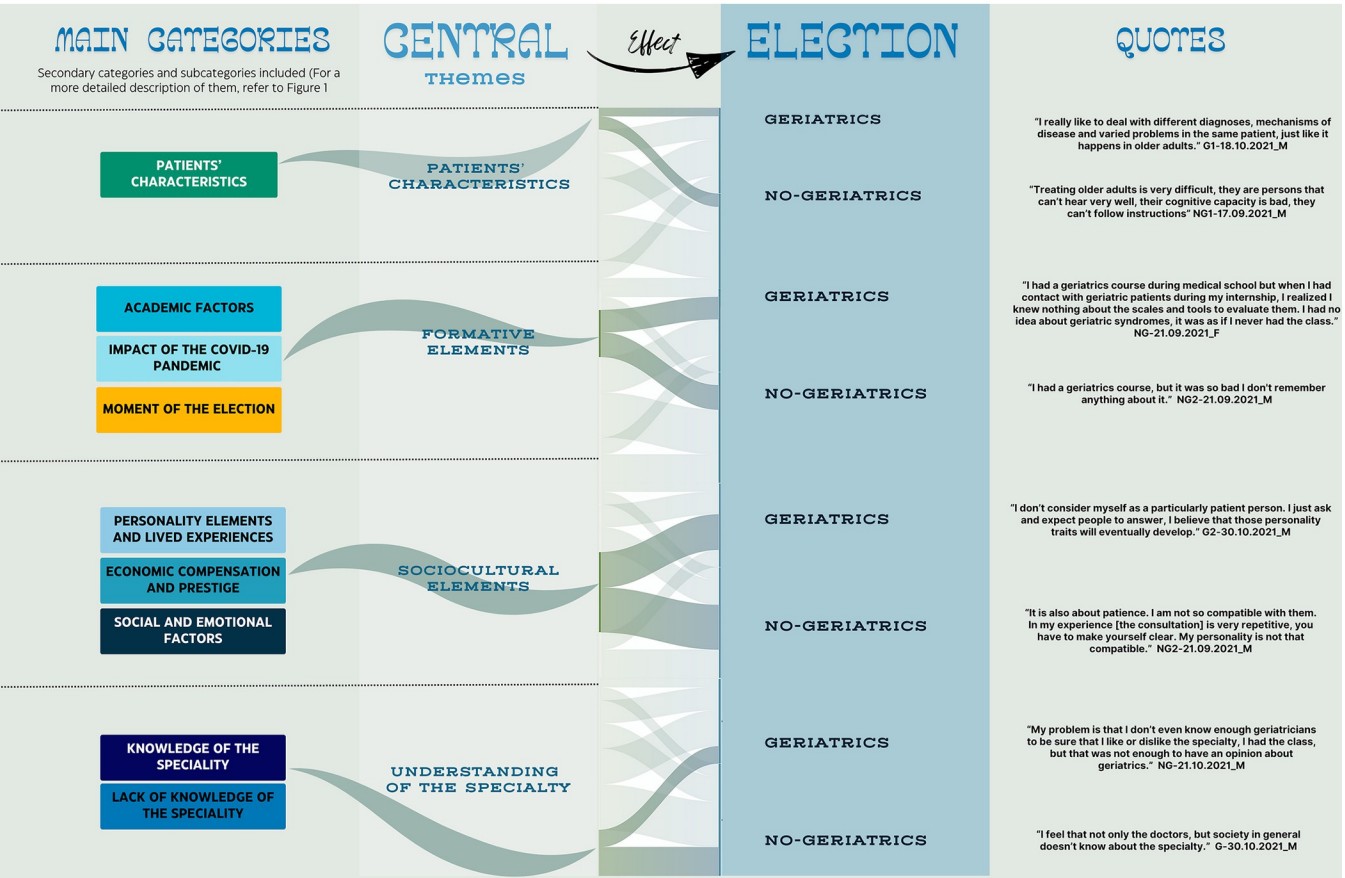

**Fig 3. Second level analysis.** Conformation of the central themes including the main and secondary categories and subcategories in the initial analysis.

CGM: geriatrics specialist, involved in medical education through a master's in health sciences education from the National Autonomous University of Mexico. To preserve the quality of the study, the main author performed a reflexive process of her personal situation as a geriatrician, collaborating with both residents and fellows as well as physicians-in-training in undergraduate scenarios. She noticed throughout her own career training to be a geriatrician, a considerable bias towards geriatrics, most apparently stemming from misconceptions from the specialty, this generated curiosity about whether or not this perceived bias was shared among other colleagues and if it impacted the selection or rejection for the subject. Conscious of her own perceptions, she actively looked to include participants with points of view which could contrast and offer a more comprehensive understanding of the situation and not only opinions which resonated with her own, this was attempted by including students with as different academic experiences as possible, while also being aware that to some extent some commonality would be encountered. We also aimed for dependency and transferability by registering in detail the characteristics of the participants, the conformation of the groups, the adjustments to the interviews, the transcription process, the labeling of the quotes and the subsequent analysis of the encountered categories and subcategories as described above and in the results section.

TVV: PhD in Health Sciences, psychologist involved in Health Sciences Education research with interest in uncertainty and burnout in physicians-in-training, grounded theory coding, among others.

CGC: internal medicine specialist, PhD in Health Sciences Education, professor for both undergraduate and postgraduate physicians in the National Autonomous University of Mexico.

For data management, the software Atlas.ti version 9.1.3 was used.

## Results and discussion

Our final sample included 34 medical students divided among the different focus groups. Six students couldn't participate due to scheduling conflicts. Demographic characteristics are described in Table 3. Women were 52% of the sample. The region with most participants was the Center region with 44% of the total. Most students were in their social service year, around 9% were in the clinical internship.

Students reported a lack of knowledge and understanding of geriatric medicine competencies, academic opportunities, and day to day practice scenarios related to geriatrics. Sociocultural and familiar elements impacted strongly in both outcomes, either rejecting or choosing geriatrics. Academic elements in particular raised concern among many of the students, because not only did this affect their selection, but generated uncertainty in their ability to treat geriatric patients. Lastly, the target population was a decisive element for many participants.

Four central themes influencing the decision emerged from the 2-step analysis: 1) Patients characteristics; 2) Understanding of the specialty, which included knowledge or lack of it; 3) Formative elements, including academic factors, impact of the COVID-19 pandemic and moment of the selection; and 4) Sociocultural elements, personality and lived experiences, economic factors, prestige and social or emotional elements were included here (Fig 3).

### Patient's characteristics

The group that rejected geriatrics made some statements that showed preconceptions about the characteristics of older adults, many of the statements incurred in generalizations or

**Table 3. Demographic characteristics of the participants.**

| | Frequency (n) | Percentage (%) |
|---|---|---|
| **Gender** | | |
| **Female** | **18** | **52** |
| **Age** | | |
| 23 | 5 | 14.70 |
| 24 | 14 | 41.17 |
| 25 | 8 | 23.52 |
| 26 | 4 | 11.76 |
| $\geq$27 | 3 | 8.82 |
| **Geographical region** | | |
| **Center** | **15** | **44.11** |
| **Western** | **5** | **14.70** |
| **Northeast** | **7** | **20.58** |
| **Northwest** | **3** | **8.82** |
| **Southeast** | **4** | **11.76** |
| **Current academic year (at the moment of the study)** | | |
| **Mandatory internship** | **3** | **8.82** |
| **Social Service (current)** | **28** | **82.35** |
| **Social Service (finished)** | **3** | **8.82** |

assumptions about older adults: *"Treating older adults is very difficult, they are persons that can't hear very well, their cognitive capacity is bad, they can't follow instructions"*, while other student said: *"Older adults are just not good patients"*.

However, the same students noted that such opinions were probably based on prejudices: *"Most of the time we have stereotypes, we think older adults may be a burden or always have a lot of diseases."* This last quote shows that the student is conscious about stereotypes affecting their perception of reality, however, some elements were correct such as a larger comorbidity load. Several quotes described objective characteristics that are associated with older adults but accompanied by opinions and reflections on their lived experiences: *"I felt that it was very sad or difficult to treat older adults, mainly those who were abandoned or with a lot of diagnoses that in the end stop being functional."*

Students that chose geriatrics had a more positive opinion about the experience of treating an older adult: *"I really like to deal with different diagnoses, mechanisms of disease and varied problems in the same patient, just like it happens in older adults."* Also, an important element was the integrality related to the study and treatment of these patients: *"Everything comes together in geriatrics, social, emotional, and medical elements. A geriatrician has to know about everything, trauma, surgery, and many other specialties, it is very interesting."* Finally, students who chose geriatrics observed the need for geriatric care in the current state of our population: *"During my social service year, the majority of the population were patients with chronic diseases and older adults."*

## Understanding of the specialty

Students in both groups consistently expressed that the available information about geriatrics is scarce. Such a lack of knowledge was identified not only in medical education settings, but also in the general population: *"I feel that not only the doctors, but society in general doesn't know about the specialty."* Also, clinical experiences during formative years were limited regarding geriatrics, for instance one of the students that rejected geriatrics said: *"I think we don't have enough contact with geriatric medicine, hospitals don't allow us to have contact with it, I mean you are aware that it exists and it's fine, but how are you supposed to like something that you don't know?"*. Students that rejected geriatrics also said that they knew very little about geriatric practice: *"My problem is that I don't even know enough geriatricians to be sure that I like or dislike the specialty, I had the class, but that was not enough to have an opinion about geriatrics."* Additionally, students that chose geriatrics noted that even when they were interested in the specialty, they had to obtain information by themselves, relying on personal effort to understand enough about geriatrics: *"I think that there is a lack of access to information about geriatrics, the information is there, but it's not like with pediatrics or OBGYN (obstetrics and gynecology) for example where information is always available"*.

## Formative elements

Regarding academic elements, students reflected that they frequently had formative experiences with classes or clinical practices that were of poor quality specially compared with other subjects: *"I had a geriatrics course, but it was so bad I don't remember anything about it."* Some students were concerned about the impact of the inferior quality of the lessons on their overall competencies: *"I had a geriatrics course during medical school but when I had contact with geriatric patients during my internship, I realized I knew nothing about the scales and tools to evaluate them. I had no idea about geriatric syndromes, it was as if I never had the class."*

This lack of quality was a source of discontent in students from both groups: *"I think that if universities are going to include geriatrics in the curriculum, they should do so properly.*

Medical schools are supposed to try to educate good general practitioners and geriatrics is a cornerstone of general medicine."

## Sociocultural elements

Sociocultural elements included personality elements, prestige of the specialty, emotional aspects and lived experiences. Students that rejected geriatrics frequently noted that personality had a major role in their selection, particularly, that their personality was 'not compatible' with the specialty as they described themselves as 'impatient' or 'desperate' and that such characteristics rendered them incompatible with geriatrics: *"It is also about patience. I am not so compatible with them. In my experience [the consultation] is very repetitive, you have to make yourself clear. My personality is not that compatible."* Many of the students in the group that rejected geriatrics shared this view of needing a particular personality in order to choose one specialty over another. This perception was not shared by the students that chose geriatrics, and some of them even described themselves as 'not patient': *"I don't consider myself as a particularly patient person. I just ask and expect people to answer, I believe that those personality traits will eventually develop."*

Previous experiences with older adults were important in both groups, either with patients or with family members, although most of them were in a medical care scenario and the experiences described were similar between the groups: family members, grandmothers or grandfathers that required care related to memory or functionality loss. The key difference among the students was the interpretation they gave to such experiences, those who rejected geriatrics perceived the experiences as stressful and emotionally challenging leading them to avoid any further contact with older adults meanwhile students that chose geriatrics felt that the experiences with older adults may be stressful, but they provided opportunities to take actions and improve the overall status of the patients.

Regarding the specialty prestige, students commented on economic remuneration and lifestyle. Students that rejected geriatrics admitted they had no particular knowledge about the lifestyle, professional opportunities or role expected from a geriatrician. Some of them said they thought geriatricians made more money than internal medicine specialists but acknowledged it was just guessing work; however, they noted that if they knew geriatricians, they usually had a high demand for consultations.

Students that chose geriatrics agreed with this as something positive and desirable: *"There is going to be a lot of work. I listened to my aunts, and they sometimes said how the geriatrician from their hometown had a full schedule for the next 4 or 5 months"*.

Lastly, students commented on the emotional impact that caring for older adults had on them as doctors. Students that rejected geriatrics felt sad when treating older adults and sometimes uneasy because they felt doctors were too aggressive with the treatment of these patients: *"It has a lot to do with quality of life and the hopes they give to many geriatric patients, this made me really uninterested because. . .maybe that would be a big emotional blow for me, seeing many of my patients very ill. . .at the verge of dying I don't think I could handle that"*. Students that chose geriatrics felt that indeed sometimes caring for older adults could be emotionally challenging, but they saw that challenge as an opportunity to look for ways to provide better care for them: *"During clinical practices in the hospital you can tell that they didn't treat patients well. I think we as doctors besides giving a diagnosis, we must try to care for them in the best possible way and not deny them anything just because they are older adults and maybe they are not going to live as long, we should provide them with quality care anyway."*

Even though at first glance specialty selection only has an impact on a personal or local level, the reality is that all of these processes have a significant effect in the conformation of the

health care services of a country. Offer-demand dynamics play an important role in this process and unlike other elements such as government policies, these dynamics are easier to overlook.

With this in mind we conducted a qualitative analysis to identify what elements have an impact in selecting or rejecting geriatrics, addressing the "demand" part of the equation. Unsurprisingly the themes and categories that we obtained added to the elements already known to impact in choosing specialty [3].

Patients' characteristics undoubtedly had a major role in the students' selection, particularly the associated image and assumptions of older patients, such as high demand of health care, complexity of diagnosis and difficult follow-up. Students who chose geriatrics seemed to enjoy the challenge of having a complex physiological and pathological scenario and having to include elements from outside the traditional medical problem approach to understand and impact the patients' situation. As expected, these same characteristics were interpreted differently by those students who rejected geriatrics, seeing them as pitfalls, instead of incentives. This finding is consistent with previous studies, which found that a specific group of patients might be of particular interest in some students [17].

In our study, previous experience, or lack thereof, was determinant in deciding regarding geriatrics, being one the most common codes found in the interviews. For those who selected geriatrics, they recognized that contact and academic content about geriatrics was very limited, and at several moments further information about the specialty was obtained due to their own efforts and in their spare time. The students recognized that the quality and quantity of geriatric content was different to that of other specialties. The importance of exposure to geriatric care has been reported before. For example, a systematic review exploring why students would not choose geriatrics, concluded that contact with the rewarding aspects of the specialty was an important element favoring students to select geriatrics [6].

Academic experiences seem to be a missed opportunity in many cases, since even the students who eventually chose geriatrics recognized that most of the time, academic contact with geriatrics was dull and of inconsistent quality.

Also, students didn't always have the opportunity to have a formal course about geriatrics and/or to take part in a rotation which included geriatric formative aspects of geriatric. As stated by the students, it is hard, if not impossible, to like something of which you are not aware.

Furthermore, previous studies have shown that positive academic and clinical experiences had a very important impact in specialty selection, which is consistent with our findings [21–24].

Students definitely can't extract themselves from the social and cultural aspects of their situation. One important item that came up in the interviews was the lived experiences and interpretation of contact as caregivers of older relatives. Students who interpreted or experienced the situation as something positive or rewarding were more likely to select geriatrics, while those who had a negative emotional impact in their life as a result of caregiving did not wish to remain in contact with older adults. It seems that there should be an interest and a connection with the patients, but the students need to be able to have emotional boundaries in order to deal with the emotional demands of caring for a particular group of patients with such special needs as the geriatric population.

Throughout the study, underlying themes emerged that linked many of the elements commented by the students which are negative perceptions and attitudes towards older adults from both them and the doctors they encountered in their daily educational activities. Such attitudes have been previously described in the literature [25] and in similar settings as our own and resonate with our findings: ageism, barriers in communication, inexperience in

geriatric care and lived experiences with older adults [25]. In fact, these findings are also consistent with attitudes registered in physicians-in-training in Mexico, as mainly negative perceptions were found towards aging and to some extent to older adults [26].

We concur that these attitudes are susceptible to educational interventions that could improve the situation regarding the perception of aging in general and geriatrics in particular among physicians-in-training.

In fact, we strongly believe that some form of "re-branding" from within the educational community for geriatrics, may increase positive attitudes towards older adults and their care, undoubtedly, even if interventions have not been universally successful, other authors have found positive effects in this regard [27].

However important work is still needed to achieve an impact in geriatrics, for instance, a study conducted by member of this research team, found that in Mexico not all the certified universities include geriatrics in their academic program and when included, the subject is imparted very heterogeneously [28], scenario that seems to be similar in different settings. Furthermore, these negative perceptions expand beyond the medical and educational communities but are also found in different aspects of current life, including social media, as shown in studies analyzing ageist content during the COVID-19 pandemic [29].

Therefore, we believe that a concerted effort must be made to change the current tendency regarding geriatrics education not only in physicians-in-training but also in graduated specialists, starting by ensuring high-quality training in geriatrics in all medical curriculum.

## Conclusions

Interestingly, students who chose geriatrics did not consider themselves with any particular personality traits, and except for patience, there were no elements consistently mentioned during the interviews. Meanwhile, students who rejected geriatrics described a very specific and detailed list of characteristics they considered medical students should have in order to be good geriatricians. Other studies have also reported this idea that you must "fit in" in a specialty in order to pursue it, and it is well documented that medical students expect to find specific personality profiles in different specialties [30]. Although personality has always been associated with a preference for certain medical specialties, the truth is that personality evaluation as a predictor for specialty selection has been inconsistent, using different questionnaires to classify specialties [31–33], and even though some associations have been found, there are no specific personality traits that can define a specialty, and even when there is a prominent personality element in a specialty there are usually other specialties in which the factor also appears [34].

Prestige and economic elements were referred as "not that important" for both groups, which is a finding that has been previously reported, very few considered economic compensation as a main determinant in their decision [35], however a more detailed evaluation of the available spots to work as a specialist in the country and how this affects specialty selection may be interesting to complement this particular aspect of the analysis.

## Study limitations

We acknowledge that further mixed-methods research should be performed in order to have a comprehensive view of specialty selection.

Even though we achieved balance in most of our groups, most students came from western and central regions of the country. Larger representation of different areas might modify or add to some of the discussed elements.

Finally, participant selection bias must be considered, particularly since all participants in our study were active in social media and we should consider that students with different points of view might have been excluded in our sample due to a lack of active engagement in social media platforms.

## Future perspectives

A larger effort to include academic and clinical aspects of geriatric care could be an effective strategy for ensuring more interest in the specialty.

Since a lot of students already don't select geriatrics as a specialty, a strategy that targets an increased exposure of medical students to core geriatric content could be effective in increasing overall interest in the specialty, as some students may have been previously indifferent towards geriatrics might develop an interest and those who reject it, would have certainly rejected it anyway.

Finally, it would be interesting to study and analyze the impact of health-care policies in different countries and how this may increase or diminish interest in a given specialty, including geriatrics, to see if these interventions may be another way to improve the specialty's standing in different settings. Also, it is recommended that medical schools in low and middle-income countries consider incorporating geriatrics into their curricula to better equip future health-care professionals with the skills and knowledge to provide quality care to the aging population [36].

## Acknowledgments

We would like to thank all the future residents from the diverse specialties who agreed to share their experiences, without them this would not have been possible.

## Author Contributions

**Conceptualization:** Carolina Gómez-Moreno, Tania Vives-Varela, Melchor Sánchez-Mendiola, Carlos Gutiérrez-Cirlos.

**Data curation:** Carolina Gómez-Moreno, Melchor Sánchez-Mendiola.

**Formal analysis:** Carolina Gómez-Moreno, Carlos Gutiérrez-Cirlos.

**Investigation:** Tania Vives-Varela, Ayari Pérez-Méndez, Carlos Gutiérrez-Cirlos.

**Methodology:** Carolina Gómez-Moreno, Tania Vives-Varela, Ayari Pérez-Méndez, Melchor Sánchez-Mendiola, Carlos Gutiérrez-Cirlos.

**Supervision:** Melchor Sánchez-Mendiola.

**Validation:** Carolina Gómez-Moreno, Ayari Pérez-Méndez, Carlos Gutiérrez-Cirlos.

**Writing – original draft:** Carolina Gómez-Moreno, Ayari Pérez-Méndez, Melchor Sánchez-Mendiola, Carlos Gutiérrez-Cirlos.

**Writing – review & editing:** Carolina Gómez-Moreno, Tania Vives-Varela, Alfonso Gulias-Herrero, José Alberto Ávila-Funes, Melchor Sánchez-Mendiola, Carlos Gutiérrez-Cirlos.

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
