## [Decision Letter · Decision Letter 0]

25 Sep 2024

PONE-D-24-29160Geriatrics as a Specialty: Insights from Mexican Medical Students' Decision-Making ProcessPLOS ONE

Dear Dr. Gutierrez-Cirlos,

Thank you for submitting your manuscript to PLOS ONE. After careful consideration, we feel that it has merit but does not fully meet PLOS ONE’s publication criteria as it currently stands. Therefore, we invite you to submit a revised version of the manuscript that addresses the points raised during the review process.

We look forward to receiving your revised manuscript.

Kind regards,

Guillermo Salinas-Escudero, PhD. MsC.

Academic Editor

PLOS ONE

Journal Requirements:

3. We note that your Data Availability Statement is currently as follows: “All relevant data are within the manuscript and its Supporting Information files.”

4. Please amend your list of authors on the manuscript to ensure that each author is linked to an affiliation. Authors’ affiliations should reflect the institution where the work was done (if authors moved subsequently, you can also list the new affiliation stating “current affiliation:….” as necessary)

Additional Editor Comments:

Despite the aging rate and health needs of this population, geriatrics is one of the least popular medical specialties. Thus, the article is relevant given the shortage of Geriatricians in Mexico and the Americas.

However, due to the characteristics of the sample, the results do not appear to be exclusive to this particular medical specialty. Specifically, the authors present among the results that the selection of the specialty is influenced by previous experiences with older people at a personal/family level, in clinical practice at the internship, or during social service. These reasons are the same for selecting other medical specialties, so what other reasons determine the higher demand for other specialties, not Geriatrics?

Many students choose their specialty mindlessly since their expectations and the reality they will face diverging regarding the caseload and working conditions.

The authors mention that the selection of the specialty has nothing to do with the economic component but with the opportunity to obtain positions that allow them to work in said specialty. However, at least in Mexico, it is almost impossible to obtain a position in the public sector today.

Indeed, many students lack a solid knowledge of the population's sociodemographic and health characteristics, which promotes less-than-objective choices. This results from the need for more content on geriatrics and other fundamental areas such as public health and epidemiology. This vicious circle condemns these specialties to undervaluation and encourages students to like other specialties that may not be what society demands.

In this sense, it would be interesting if the authors presented an analysis of these programs' shortcomings and areas of opportunity as part of their discussions. Likewise, it is necessary to delve into the possible mechanisms by which universities change the academic content based on the real needs of the population and, therefore, promote this specialty during undergraduate training.

Another limitation observed is that, although the sample represents students by region, the authors are not clear about whether public and private universities were included equally and in a balanced way.

It would be interesting to conduct the same study on a sample of graduated specialists and determine the differences in perspectives.

Reviewers' comments:

Reviewer's Responses to Questions

**Comments to the Author**

1. Is the manuscript technically sound, and do the data support the conclusions?

Reviewer #1: Partly

Reviewer #2: Yes

2. Has the statistical analysis been performed appropriately and rigorously? 

Reviewer #1: N/A

Reviewer #2: I Don't Know

3. Have the authors made all data underlying the findings in their manuscript fully available?

Reviewer #1: Yes

Reviewer #2: No

4. Is the manuscript presented in an intelligible fashion and written in standard English?

Reviewer #1: Yes

Reviewer #2: Yes

5. Review Comments to the Author

Reviewer #1: The study offers valuable insights into why Mexican medical students choose or reject geriatrics as a specialty. However, the manuscript could be improved by providing a more detailed explanation of the coding process and validation steps to enhance the rigor of the qualitative analysis. Including a more in-depth discussion about the implications of these findings for medical education could also add value. The use of quotes to illustrate themes is effective, but a clearer link between specific quotes and the themes/categories would provide stronger support for the findings.

Reviewer #2: This study provides relevant insight into the factors that drive medical student towards considering or rejecting geriatric medicine as a career. It is surprising that stereotypes or preconceptions about aging, old age and older persons were not addressed in the discussion, although they indeed seem to emerge during the focus groups. Stereotypes might play a role in shaping students' opinion on what it is like to work with older persons, especially in the abscence of previous actual experience with health care of older persons. This has been well explored in the literature (Nojomi, M., Goharinezhad, S., Saraei, R. et al. Exploring the attitudes of general medical students toward older adult’s care in a lower middle-income country: implications for medical education. BMC Med Educ 23, 649 (2023). https://doi.org/10.1186/s12909-023-04626-1). Even short, knowledge-building interventions may change medical students' and physicians' attitudes towards older persons (Samra R, Griffiths A, Cox T, Conroy S, Knight A. Changes in medical student and doctor attitudes toward older adults after an intervention: a systematic review. J Am Geriatr Soc. 2013 Jul;61(7):1188-96. doi: 10.1111/jgs.12312. Epub 2013 Jun 10. PMID: 23750821; PMCID: PMC3808566.) It would be interesting to know the authors' take on how stereotypes may underly in the decision-making in their study.

On the other hand, it would be interesting to acknowlege how non-tangible environmental factors may play a role in the medical students' decision to consider or reject geriatric medicine as a career option. Student awareness of local and national policies on aging, caregiving and health care may play a role. I am aware that this might not be possible to addres in the current paper but it should be acknowledged as part of future directions.

6. PLOS authors have the option to publish the peer review history of their article (what does this mean?). If published, this will include your full peer review and any attached files.

Reviewer #1: No

Reviewer #2: No

---

## [Author Response · Author response to Decision Letter 0]

18 Nov 2024

Mexico City, November 18th, 2024

Solna Carreon Santos

Editor

PLOS ONE

We are submitting a revised version of our manuscript: “Geriatrics as a Specialty: Insights from Mexican Medical Students' Decision-Making Process” PONE-D-24-29160. According to your letter of November 18th, 2024:

“Please ensure that you include a title page within your main document. You should list all authors and all affiliations as per our author instructions and clearly indicate the corresponding author”.

According to your instructions:

- The title page was included in the main manuscript.

- The list of authors was modified according to the instructions for authors, institutional positions and academic titles were eliminated and the symbols indicated in the instructions were used.

The corresponding author was specified according to the instructions:

* Corresponding author

alfonso.gutierrezc@incmnsz.mx
cirlos@hotmail.com (CGC)

Thank you very much for your quick response and patience.

Kind regards.

Carlos Gutiérrez-Cirlos

Faculty of Medicine, National Autonomous University of Mexico, Mexico City, Mexico.

---

## [Editor Report · Decision Letter 1]

21 Nov 2024

Geriatrics as a Specialty: Insights from Mexican Medical Students' Decision-Making Process

PONE-D-24-29160R1

Dear Dr. Gutierrez-Cirlos,

We’re pleased to inform you that your manuscript has been judged scientifically suitable for publication and will be formally accepted for publication once it meets all outstanding technical requirements.

Kind regards,

Guillermo Salinas-Escudero, PhD. MsC.

Academic Editor

PLOS ONE

Additional Editor Comments (optional):

This new version of the work has the quality requirements to be published
---

## [Editor Report · Acceptance letter]

26 Nov 2024

PONE-D-24-29160R1 

PLOS ONE

Dear Dr. Gutierrez-Cirlos, 

I'm pleased to inform you that your manuscript has been deemed suitable for publication in PLOS ONE. Congratulations! Your manuscript is now being handed over to our production team.

Kind regards, 

on behalf of

Dr. Guillermo Salinas-Escudero 

Academic Editor

PLOS ONE